# Effect of Replacing Corn Grain and Soybean Meal with Field Peas at Different Levels on Feed Intake, Milk Production, and Metabolism in Dairy Cows under a Restrictive Grazing

**DOI:** 10.3390/ani14192830

**Published:** 2024-09-30

**Authors:** Ruben G. Pulido, Ignacio E. Beltran, Jorge A. Aleixo, Álvaro G. Morales, Marcelo Gutierrez, Matias Ponce, Pedro Melendez

**Affiliations:** 1Instituto de Ciencia Animal, Facultad de Ciencias Veterinarias, Universidad Austral de Chile, Valdivia P.O. Box 567, Chile; aleixoantonio36@gmail.com (J.A.A.); alvaro.morales@uach.cl (Á.G.M.); 2Instituto de Investigaciones Agropecuarias, INIA Remehue, Osorno P.O. Box 24-O, Chile; ignacio.beltran@inia.cl; 3Graduate School, Faculty of Veterinary Sciences, Universidad Austral de Chile, Valdivia P.O. Box 567, Chile; 4ALISUR S.A., Ruta U 145 Km 0.7 S/N, San Pablo P.O. Box 1040, Chile; marcelo@trebun.cl; 5School of Veterinary Sciences, Faculty of Veterinary and Livestock Sciences, Universidad de Chile, Santiago P.O. Box 1004, Chile; matias.ponce.c@ug.uchile.cl; 6School of Veterinary Medicine, Texas Tech University, Amarillo, TX 79106, USA; pedro.melendez@ttu.edu

**Keywords:** pea-based concentrate, grazing, dairy cattle, sustainability

## Abstract

**Simple Summary:**

In pasture-based dairy systems, supplementing the diet is crucial during periods when pasture availability is low. Field peas, which can be locally produced, offer relatively high concentrations of crude protein, along with significant energy content in the form of starch, all at a lower price. These qualities make field peas a sustainable alternative to traditional protein and energy sources such as soybean meal and corn grain. This study explored the impact of substituting traditional feeds with field peas in the concentrates of grazing dairy cows. Our findings indicate that using field peas maintains milk production and composition (protein and fat) without affecting intake or ruminal and blood parameters. This suggests that field peas are a viable alternative for feeding dairy cows, supporting consistent production under grazing conditions.

**Abstract:**

This study assessed the effects of replacing soybean meal (SBM) and corn grain with field peas in the concentrate of grazing dairy cows on milk production, intake, ruminal fermentation, and blood indicators. Twelve multiparous lactating Holstein-Friesian cows were utilized in a replicated 3 × 3 Latin square design, comprising three periods and three treatments: (1) Pea-0 (Control diet): 6 kg dry matter (DM) of fresh pasture, 7.2 kg DM of grass silage, and 7 kg DM of a concentrate containing 0% pea; (2) Pea-30: Control diet with the concentrate composed of 30% pea; (3) Pea-60: Control diet with the concentrate composed of 60% pea. The effect of treatments on productive and metabolic parameters was evaluated using linear-mixed models. Pasture and total DM intake, milk production, and composition were unaffected by treatments. Despite the concentrates being isonitrogenous and isoenergetic, crude protein (CP) intake was slightly higher in Pea-30 and significantly higher in Pea-60 due to higher pasture CP content in the pasture grazed by these groups, leading to higher milk urea content, though within recommended ranges. Blood parameters showed no significant changes, except for plasma β-hydroxybutyrate, which was lowest in the Pea-60 treatment; however, all values were within ranges not indicative of subclinical ketosis. Ruminal fermentation parameters were similar across treatments. These findings support the use of field peas as a viable alternative to replace SBM and corn grain in concentrates, enabling similar milk production and composition in grazing dairy cows.

## 1. Introduction

In pasture-based dairy systems, productivity and profitability can be greatly improved by increasing the quantity, quality, and efficient utilization of growing pastures [1]. Spring and fall in temperate regions are generally times of accelerated growth rates with high nutritive value, while summer is a time of reduced growth, lower forage yield, and quality [2]. Consequently, adjustment is needed for the diets of the cows under grazing conditions since seasonal variations in pasture quality and quantity have been a challenge in the last years [3]. Restricting access to pasture when pasture growing is low can be used as a management tool to increase grazing efficiency and pasture utilization [4,5]. However, restricting access time to pastures reduces herbage dry matter intake (DMI) and milk production by dairy cows [6]. Therefore, it is essential to provide supplementary feeds to meet the nutritional requirements of the milking herd [7]. Traditional strategies for feeding supplements to grazing cows involve concentrates fed in the milking parlor, complemented with forages such as silages, forage crops, and by-products fed as a partial mixed ration (PMR) before or after milking the herd.

Soybean meal (SBM) and corn grain are commonly used ingredients of concentrates for dairy cows in many parts of the world, providing high amounts of metabolizable protein and energy, becoming a key component of feeding programs. The use of alternative sources of plant protein to soybean meal in diets for agricultural animals aims to reduce soybean imports into the EU and countries like Chile and partially substitute genetically modified organisms in the food chain [8,9]. This could help to reduce the carbon footprint of dairy cattle, as soy is recognized to have a high carbon footprint (4.152 kg CO_2eq_/kg product; [10]) compared to other protein sources such as peas (0.49 kg CO_2eq_/kg product; [11]). Consequently, employing local protein sources for dairy cattle is viewed as an effective strategy to reduce both the carbon footprint and the need for food importation, ultimately lessening the dependency on international supplement supplies.

Field peas (*Pisum sativum*), also known as dry peas, are available in various forms such as raw, split, ground, and extruded, among others. They are among the few feedstuffs that offer relatively high concentrations of crude protein (CP) and well-balanced essential amino acids, along with significant energy content in the form of starch, all at a lower price [12,13]. These characteristics make field peas an effective alternative to traditional protein and energy sources such as SBM, corn grain, and barley in dairy cow rations [14], thus supporting more locally sourced and environmentally friendly farming practices.

Replacing SBM and corn grain with proteaginous seeds in dairy cow diets has been studied for many years [15,16,17]. However, field studies that specifically utilize peas as the primary source of protein and starch for grazing dairy cows are scarce. Inconsistent results have been reported in the literature, with no effect on DMI and milk production [18,19] or reduction in DMI and a linear decrease in milk production and protein yield [20]. It is well-known that most of the rumen degradable protein (RDP) in field peas is in the form of soluble protein (SP), which may decrease milk yield and N utilization in lactating dairy cows [15]. Albrecht [20] reported that these negative responses were more severe at high levels of inclusion of cracked field peas (≥24% of diet DM) at the expense of ground corn and SBM, or dairy cows with a high genetic merit for production, thereby suggesting that depressed DMI and deficiency in the supply of metabolizable lysine (MP-Lys) or methionine (MP-Met) or both might be involved [14]. Previous studies [18,19,21] conducted with lactating dairy cows showed inconsistent results in the ruminal concentration of total VFA and molar proportion of individual VFA when replacing SBM and corn or barley with field peas (ground, raw, extruded, or expanded) [14]. These variations indicate that differences in diet composition, the inclusion levels of field peas, the method of grain processing, and specific animal requirements likely influence the response of the animals. Considering the potential environmental and economic benefits of using field peas, as highlighted earlier, this study hypothesizes that replacing soybean meal with field peas in the diets of dairy cows in pasture-based systems during summer—a season characterized by restrictive pasture conditions and reduced herbage quality—will not only meet the nutritional needs of the cows but also enhance sustainability and reduce costs. Therefore, the objective of this study was to quantify the effects of replacing SBM and corn grain with field peas at various substitution levels in the concentrate on animal performance, ruminal function, and blood indicators in grazing dairy cows during the summer.

## 2. Materials and Methods

The experiment was carried out between 26 January and 30 March 2020, at the Austral Agricultural Research Station of the Universidad Austral de Chile (39°47′ S, 73°l4′ W). The animal handling procedures described in this experiment were approved by the Animal Welfare Committee of the Universidad Austral de Chile (Protocol number C28-2020).

### 2.1. Animals and Experimental Design

Twelve multiparous lactating ruminally fistulated Holstein-Friesian cows were used in a replicated 3 × 3 Latin square, with three treatments and three periods, where animals allocated to each treatment were used as experimental unit. Each experimental period had a 21 d duration. The first 14 days of each period were considered as the adaptation time to the diets, and the last 7 days as the experimental period for sample collection. Cows were assigned to treatments and blocked by milk production (25.6 ± 2.0 kg/d), body weight (BW; 641± 21.8 kg), and days in milk (DIM; 152 ± 33.4). They were randomly allocated to one of three treatments (*n* = 4 cows per treatment): (1) Control diet: 6 kg DM of fresh ryegrass pasture (*Lolium perenne*), 7.2 kg DM of ryegrass silage, 7 kg concentrate (Pea-0: 100% concentrate mixture with 0% of ground pea), and 0.21 kg of mineral/vitamin premix; (2) Pea-30: 6 kg DM of fresh ryegrass pasture, 7.2 kg DM of ryegrass silage, 7 kg concentrate (Pea-30: 30% ground pea in the concentrate mixture), and 0.21 kg of mineral/vitamin premix; and (3) Pea-60: 6 kg DM of fresh ryegrass pasture, 7.2 kg DM of ryegrass silage, 7 kg concentrate (Pea-60; 60% ground pea in the concentrate mixture), and 0.21 kg of mineral/vitamin premix. The concentrates, including all three formulations (Pea-0, Pea-30, and Pea-60), were manufactured and supplied by ALISUR S.A. (San Pablo, Región de Los Lagos, Chile). Each formulation was prepared to be isonitrogenous and isoenergetic, containing 21% CP and 3.2 Mcal of metabolizable energy (ME)/kg DM. Of the total 7 kg of concentrate designated per cow per day, only 1 kg was divided and individually offered during the milking sessions—0.5 kg during the morning milking and 0.5 kg during the afternoon milking. The remaining 6 kg of concentrate, along with all the silage and mineral salts, was provided as a PMR in individual feeding pens in the cowshed after afternoon milking. Water was available via free-access troughs in the paddock, and in the cowshed, cows had access to individual automatic waterers in their stalls. A summary of the ingredients used in formulating the three types of concentrates is presented in Table 1, while the chemical composition and ingredient percentages of these formulated concentrates are detailed in Table 2.

### 2.2. Grazing Management

The experiment involved rotational strip-grazing across three paddocks, collectively covering a total area of approximately 10 hectares. This rotational system allowed each treatment group to graze one paddock at a time, moving to the next when the available forage was consumed, and returning to the first paddock once regrowth was sufficient to support grazing again. Each paddock was divided into strips, separated by electric fences to ensure that each treatment grazed independently.

Each cow was provided with a daily pasture allowance of 15 kg DM cow/d, measured at ground level. This allowance was allocated daily after the morning milking at 09:00 h, allowing cows to graze for 8 h. The daily grazing area for each treatment was adjusted based on herbage allowance and pre-grazing herbage mass (HM, kg DM/ha, measured above ground level).

Pre-grazing HM was estimated using a Rising Plate Meter (RPM; Ash-grove Plate Meter, Hamilton, New Zealand). For each grazing area assigned to a treatment, a total of 100 RPM measurements were taken. Herbage mass was then calculated using a specific equation formulated for spring pastures in southern Chile: HM, kg DM/ha = RPM × 100 + 400; R2 = 0.75 [22]. Post-grazing HM was assessed using the same methodology. It is crucial to note that the treatments grazed through the paddocks in a consistent order: Pea-0 on the left, Pea-30 in the middle, and Pea-60 on the right.

### 2.3. Pasture and Supplements

The pasture utilized for the experiment, primarily composed of *Lolium perenne* and maintained under irrigation, was organized into 50 m wide paddocks. These were further subdivided into strips ranging from 100 to 500 m from the milking parlor to facilitate efficient grazing management. Grass silage was produced from pastures at the University Agricultural Research Station. Briefly, herbage was cut down and then withered for 24 h before being stored and preserved in plastic bales.

Herbage samples were taken thrice during each experimental period just before the cows were introduced to a new strip at 08:30 h, and all samples were cut at 4 cm above ground level. Additionally, grass silage samples were collected once per experimental period.

All collected samples were immediately frozen to maintain their integrity until analysis could be performed. In the Animal Nutrition Laboratory at Universidad Austral de Chile, the samples were freeze-dried and ground through a 1 mm sieve (Wiley Mill, 158 Arthur H. Thomas, Philadelphia, PA, USA). They were then analyzed for DM, CP, acid detergent fiber (ADF), ash and lipid [23], neutral detergent fiber (NDF) [24], and ME [25]. Water-soluble carbohydrates (WSC) were determined using near-infrared spectroscopy (R^2^ of equation = 0.97). Starch quantification was conducted by colorimetric detection of non-soluble residues after enzymatic digestion with amylglucosidase according to Pelletier et al. [26].

### 2.4. Dry Matter Intake

Herbage DMI was estimated using the methodology described by Al-Marashdeh et al. [27]. It was determined on days 3, 5, and 7 of week 3 of each experimental period. To estimate herbage DMI, cows were paired according to their respective experimental square, and a single strip was offered to each pair. Herbage DMI for each pair was calculated by subtracting the post-grazing herbage mass from the pre-grazing herbage mass and then dividing by the number of cows in each subgroup.

Additionally, the amounts of PMR offered and the refusals were recorded daily during the last week of each experimental period to estimate individual supplemental DMI. Total DMI was calculated by summing the intakes from both pasture and PMR.

### 2.5. Milk Production and Body Weight

Cows were milked at 07:00 and 15:00 h, and milk yields were recorded daily using a flow sensor (MPC580; DeLaval, Tumba, Sweden) during the experimental periods. The daily average milk yield for the final week of each period is reported. Milk samples were collected during the last week of each experimental period from two consecutive milkings and were analyzed separately for protein, fat, and urea content using infrared spectroscopy (Milko-scan, System 4300, Foss Electric, Hillerød, Denmark). Body weight was recorded daily after each milking session using an automated system in the milking parlor (DeLaval, Tumba, Sweden).

### 2.6. Blood and Ruminal Parameters

Blood samples were collected at 15:30 and 21:30 h on one day during the last week of each experimental period. Samples were drawn from the coccygeal vessels using heparinized vacutainers and immediately transported on ice to the clinical pathology laboratory of the Universidad Austral de Chile. In the laboratory, blood samples were centrifuged at 3000× *g* for 10 min to obtain plasma, which was then frozen at –20 °C for subsequent analysis. Plasma was analyzed for β-hydroxybutyrate (BHB) (Ranbut; Randox Laboratories, Crumlin, County Antrim, Northern Ireland), urea (glutamate dehydrogenase, HUMAN, Wiesbaden, Germany), albumin, calcium (Ca-color Arsenazo III), phosphorus (Phosphorus Liquirapid [Human]), and the enzymatic activity of Aspartate Aminotransferase (AST) using a Wiener Metrolab 2300^®^ autoanalyzer (Wiener Laboratory, Rosario, Argentina). Magnesium concentration was determined using atomic absorption spectrophotometry (Thermo Scientific, Waltham, MA, USA).

Ruminal fluid samples (20 mL) were collected at the same time as the blood samples using a stomach tube (Flora Rumen Scoop; Prof-Products, Guelph, ON, Canada). Samples were strained through four layers of cheesecloth, and 10 mL of ruminal fluid was mixed with 0.2 mL of 50% (*w*/*v*) sulfuric acid and stored at −20 °C until laboratory analysis. For analysis, samples were thawed for 16 h to 4 °C, then centrifuged at 10,000× *g* for 10 min at 4 °C. Six milliliters of the supernatant were drawn off and further centrifuged under the same conditions. The thawed supernatant of the ruminal fluid samples was analyzed for volatile fatty acids (VFA) by gas chromatography (GC; Shimadzu GC-2010 Plus High-end gas chromatography, equipped with a capillary column SGE, BP21 (FFAP); Shimadzu Corporation, Kyoto, Japan), as described by Tavendale et al. [28].

### 2.7. Statistical Analysis

The effects of treatments on milk production, milk composition, BW, ruminal determinations, and blood parameters were analyzed using PROC MIXED in SAS (SAS Institute Inc., Cary, NC, USA). The statistical model incorporated the fixed effects of treatment and period, and the random effects of squares and cows nested within squares. For ruminal VFA, the hour of sampling was treated as a repeated measure, and the interaction between treatment and time was tested. The degrees of freedom method employed was Kenward–Roger. The variance-covariance structure was selected based on the lowest corrected Akaike Information Criterion, which indicated the best fit for the final models. 

When the effect of treatments was significant (*p* < 0.05), orthogonal polynomial contrasts were used to determine linear and quadratic effects of the treatments on the evaluated variables. All data adhered to the assumptions of normality and homogeneity of variance. Comparisons between treatments were conducted using the Tukey test. Results were considered statistically significant at *p* < 0.05 and indicative of a tendency at *p* ≤ 0.1.

## 3. Results

### 3.1. Chemical Composition of Forages

The chemical composition of pasture and pasture silage is presented in Table 3. The chemical composition of pasture was similar across treatments (*p* > 0.05) for DM, ash, NDF, ADF, EE, starch, and WSC, averaging 30%, 9.0%, 46%, 30%, 3.0%, 3.0%, and 14%, respectively. The crude protein content and ME of the pasture were greater for Pea-60 compared to Pea-30 and Pea-0 (*p* ≤ 0.05). Grass silage presented an average of 36% DM, 13.2% CP, and 2.26 Mcal/kg DM.

### 3.2. Grazing Management and DMI

Results of grazing management and DMI are presented in Table 4. Pre-grazing and post-grazing HM were similar among treatments (*p* > 0.05), averaging 2369 and 1751 kg DM/ha, respectively. The intake of pasture, grass silage, and concentrate were not affected by the treatments (*p* > 0.05), averaging 3.9, 7.1, and 6.2 kg DM/d, respectively. Consequently, total DMI was similar across treatments, averaging 17.2 kg DM/d. Although total DMI did not vary by treatment, CP intake was significantly higher in the Pea-60 treatment compared to Pea-0 (0.14 kg CP/d), showing an approximate increase of 5% (*p* = 0.04). The average CP percentages of the diets were 16.5, 16.8, and 17.2% for Pea-0, Pea-30, and Pea-60, respectively. Intake of NDF and ME remained similar across treatments (*p* > 0.05).

### 3.3. Milk Production and Body Weight

Results of milk production and body weight are presented in Table 5. Milk production, energy-corrected milk, and fat- and protein-corrected milk did not differ among treatments, averaging 23.9 kg/d, 27.3 kg/d, and 25.6 kg/d, respectively. Similarly, fat and protein in milk (as % and kg/d) were not modified by treatments (*p* > 0.05). Urea content in milk was significantly influenced by treatments (*p* = 0.02), being 15.8% higher in Pea-30 and 17.5% higher in Pea-60 compared to Pea-0. Body weight and change in body weight during this time did not differ across treatments (*p* > 0.05).

### 3.4. Blood and Ruminal Parameters

The results of the blood parameters are presented in Table 6. Plasma concentrations of BHB were significantly modified by treatments (*p* = 0.03), showing a linear effect where BHB decreased with higher pea inclusion in the concentrate, being lowest in the Pea-60 treatment compared to Pea-0 and Pea-30. Concentrations of urea, albumin, and AST did not differ among treatments (*p* > 0.05). Similarly, mineral levels (total Ca, P, and Mg) were not modified by treatments, averaging 2.3, 1.6, and 0.8 mmol/L, respectively. 

Results of ruminal fermentation parameters are presented in Table 7. Ruminal pH and total VFA concentrations did not differ among treatments. A significant effect of sampling time (hour of sampling) was observed on the ruminal concentrations of butyrate and propionate (*p* < 0.01), which were 21% and 4.0% higher at night compared to afternoon sampling, respectively. There was no significant interaction between treatment and day (*p* > 0.05).

## 4. Discussion

Several studies have explored the effects of incorporating field peas in concentrates as a replacement for cereal grains on the productive and metabolic parameters of dairy cows fed a total mixed ration [14,19,31]. To the knowledge of the authors, this study is the first to evaluate the effects of a pea-based concentrate supplemented with grazing dairy cows with restricted access to pasture on milk production, DMI, blood metabolites, and ruminal fermentation parameters.

### 4.1. Dry Matter and Nutrient Intake

Replacing cereal grains with field peas in concentrates did not affect the herbage and total DMI. Similar findings have been documented in the literature, demonstrating that cows fed pea-based concentrates as replacements for SBM and barley [18], as well as ground corn and SBM [14], exhibited comparable apparent total-tract digestibility of NDF and ADF [14]. These observations suggest that the inclusion of peas in the concentrate maintains similar total-tract DM digestibility, thereby resulting in consistent total DMI.

Despite no modifications in DMI by treatments, CP intake was greater for the Pea-60 treatment compared to Pea-0, which corresponded with the higher CP content of the pasture offered to those animals. Similarly, a significant difference was noted in the ME content of the pastures offered to different treatments; however, ME consumption remained similar across treatments. Although cows grazed in adjacent strips within the same paddocks, small variations in the quality of pasture (CP and ME) were observed. This variation was due to the treatments grazing through the paddocks in a consistent order throughout this study. We recognize a potential limitation in this study due to the additional provision of 0.14 kg CP/cow/d in the Pea-60 treatment, which could potentially mask any nitrogen deficiency in this treatment. However, the CP values of the diet are within the recommended levels for dairy cows [13], aligning with the blood urea levels as will be discussed later. Interestingly, this extra 5% CP contribution to the Pea-60 treatment might have been excessive, as it resulted in increased blood urea levels. Given that the objective of this study was to evaluate the replacement of soybean meal with field peas in concentrates, this discrepancy would not affect our primary goal, and this is supported by the fact that no differences were observed in milk production or composition, as will be discussed later.

### 4.2. Milk Production and Composition

Milk production was not affected by the inclusion of field peas in the diet, a finding that aligns with the similar DMI observed among treatments. Dry matter intake is a key determinant of milk yield under both grazing conditions [32,33] and concentrate-based dairy systems [34]. This aligns with reports in the literature where replacing lupin [31] or ground corn and SBM [14,19] with field peas did not alter milk production in lactating dairy cows. However, Albrecht [20] and Pereira et al. [14] noted negative production responses at high levels of cracked field peas inclusion (≥24% of diet DM) or with high-producing dairy cows, likely due to DMI depression or a deficiency in the supply of MP-Lys and/or MP-Met.

The lack of significant differences in milk production across treatments might reflect an adequate microbial protein supply supporting the moderate levels of milk yield observed in this study [35]. This suggests that the quantity and quality of supplemented rumen-undegradable protein (RUP) had a minimal impact on milk production. Nonetheless, it remains uncertain if any amino acid deficiencies could potentially limit milk production in this study.

Milk fat and protein concentrations were consistent across treatments, aligning with the findings of Vander Pol et al. [19] and Tufarelli et al. [36], who observed no significant differences when replacing corn grain and soybean meal with field peas. Similarly, Corbett et al. [37] found no significant differences in milk composition when soybean and canola meal were substituted with field peas in the diets of dairy cows. Furthermore, similar results have been reported in other species, such as buffalo and sheep, where the replacement of soybean meal with field peas did not adversely affect milk yield and composition, supporting these findings [38,39].

Milk urea content was greater for treatments supplemented with peas (Pea-30 and Pea-60), likely due to their increased N intake, a critical factor influencing milk urea content [40]. Previous research has demonstrated a linear relationship between nitrogen intake and milk urea content [41,42]. It is suggested that the ruminally available N exceeded microbial requirements for protein synthesis in treatments receiving a pea-based concentrate. This excess was likely due to the similar concentration of ME in the diet, resulting in an increased production of ammonia in the rumen. This ammonia was then absorbed and converted to urea in the liver [43]. Recommended milk urea concentrations for dairy cows typically range from 250 to 350 mg/L. Our findings align with this range, despite physiological variations that may allow for higher values [44]. 

Thus, field peas can effectively replace corn grain and SBM in the diet, maintaining milk production and composition (milk protein and fat). The observed increase in milk urea content for grazing dairy cows was primarily attributed to variations in pasture quality rather than to the pea supplementation itself. 

### 4.3. Blood and Ruminal Parameters

Blood parameters remained consistent across treatments except for the plasma concentration of BHB. This ketone body is utilized to assess the energy balance in early lactation dairy cows, where an excessive mobilization of adipose reserves leads to increased levels of non-esterified fatty acids (NEFA) and ketone bodies (acetone, acetoacetate, and BHB) in the blood [45,46]. The lower plasma concentration of BHB in the Pea-60 treatment suggests a comparatively better energy balance. Despite this variation, all treatments maintained BHB concentrations below the threshold indicative of subclinical ketosis (>1.2 mmol/L) [45,46]. Given that the formulated concentrates were isoenergetic, no differences in ME intake were expected, aligning with the observed uniformity in performance outcomes. Additionally, the energy supply appeared to be limited across all diets, as evidenced by the negative body weight changes observed in all treatments during the study periods.

According to Dieho et al. [47], concentrations of VFA in the rumen increase with the intake of DM and rumen-fermentable organic matter following the onset of lactation. In this study, varying levels of peas in the concentrate did not influence the concentrations of total VFA, acetate, propionate, butyrate, isobutyrate, and valerate. This lack of effect on VFA concentrations could be attributed to consistent DMI and chemical composition of the diets across all treatments, suggesting that similar VFA concentrations are indicative of uniform ruminal fermentation patterns. Additionally, the rumen pH was consistently high with only minor variations among treatments, likely due to the similar VFA profiles observed.

Ruminal concentrations of butyrate, propionate, and valeric were greater during the night sampling compared to the afternoon, which may be associated with normal ruminal fermentation patterns, especially in cows consuming silage and concentrate as a PMR. Wang et al. [48] observed that VFA concentrations were significantly affected by sampling time, increasing post-feeding compared to pre-feeding. This supports our findings, indicating a typical fluctuation in VFA levels in response to feeding schedules.

## 5. Conclusions

The current results indicate that field peas can effectively replace corn grain and soybean meal in the diets of grazing dairy cows, maintaining similar milk production and composition (protein and fat) without altering dry matter intake or ruminal parameters. Blood β-hydroxybutyrate concentrations were slightly modified but remained within normal ranges across all treatments, indicating no adverse energy balance. The observed increase in milk urea was linked to variations in pasture quality rather than to pea supplementation. Finally, an economic analysis is recommended to determine if replacing corn grain and SBM with field peas is cost-effective.

## Figures and Tables

**Table 1 animals-14-02830-t001:** Chemical composition of ingredients used in formulating concentrates.

Chemical Composition ^1^	Field Pea	Soybean Meal	Triticale	Corn Grain	Wheat Brand
Nutrient, % DM					
Ash, %	5.35	6.00	2.36	1.31	5.64
CP, %	24.9	51.3	10.2	8.00	16.8
NDF, %	7.01	8.90	11.5	7.75	38.8
ADF, %	4.20	5.50	3.02	3.20	15.5
EE, %	1.95	1.52	1.86	3.74	4.10
CF, %	4.09	4.52	2.70	2.00	10.5
NFC, %	60.8	32.3	74.1	79.2	34.6
ME, Mcal/kg DM	3.39	3.15	3.32	3.44	2.66

^1^ Abbreviations: DM = dry matter; CP = crude protein; NDF = neutral detergent fiber; ADF = acid detergent fiber; EE = ether extract; CF = crude fiber; NFC = non-fiber carbohydrates; ME = metabolizable energy.

**Table 2 animals-14-02830-t002:** Chemical composition and ingredient percentages of the three formulated concentrates.

Chemical Composition ^1^	Treatments		
	Pea-0	Pea-30	Pea-60
Nutrient, % DM			
CP	21.0	21.0	21.0
NDF	14.3	13.8	14.3
ADF	7.05	6.92	6.98
EE	2.47	2.43	2.71
ME, Mcal/kg DM	3.20	3.20	3.20
Ash	5.88	5.39	4.96
Starch	48.2	46.7	45.1
NFC	58.3	58.0	57.4
Concentrate composition, % DM			
Triticale	35.0	35.0	—
Soybean meal	18.0	6.0	—
Wheat bran	14.0	18.0	27.4
Corn grain	32.0	10.0	12.1
Field pea	—	30.0	60.0
Urea	1.0	1.0	0.5

^1^ Abbreviations: Pea-0 = concentrate with 0% of pea; Pea-30 = concentrate composed of 30% of pea; Pea-60 = concentrate composed of 60% of pea. DM = dry matter; CP = crude protein; NDF = neutral detergent fiber; ADF = acid detergent fiber; EE = ether extract; ME = metabolizable energy; NFC = non-fiber carbohydrates; — denotes feeds that were replaced in the different concentrate formulations.

**Table 3 animals-14-02830-t003:** Chemical composition of pasture and pasture silage offered to grazing dairy cows during the experiment.

Chemical Composition ^1^	Pasture Composition per Treatment			SEM ^2^	*p*	Grass Silage	SD ^3^
	Pea-0	Pea-30	Pea-60				
DM, %	31.1	30.0	29.0	2.03	0.79	36.8	4.74
Ash, %	8.59	8.56	9.18	0.20	0.23	7.69	1.52
CP, %	15.6 ^a^	17.0 ^b^	17.6 ^c^	0.07	<0.01	13.2	0.07
NDF, %	47.3	45.5	45.0	0.94	0.31	50.2	1.66
ADF, %	31.5	30.0	29.7	0.54	0.06	33.8	2.98
EE, %	2.86	2.95	2.87	0.04	0.42	2.81	0.41
Starch, %	2.97	3.06	2.86	0.08	0.25	2.30	1.06
WSC, %	14.1	14.0	14.2	0.63	0.64	7.37	1.52
ME, Mcal/kg DM	2.36 ^a^	2.39 ^b^	2.43 ^c^	0.01	<0.01	2.26	0.04

^1^ Abbreviations: Pea-0 = concentrate with 0% of pea; Pea-30 = concentrate composed of 30% of pea; Pea-60 = concentrate composed of 60% of pea. DM = dry matter; CP = crude protein; NDF = neutral detergent fiber; ADF = acid detergent fiber; EE = ether extract; WSC = water-soluble carbohydrates; ME = metabolizable energy. ^2^ Standard error of the mean. ^3^ Standard deviation. ^a–c^ Values within a row with different superscripts differ significantly at *p* < 0.05.

**Table 4 animals-14-02830-t004:** Grazing management and dry matter intake (DMI) of grazing dairy cows supplemented with concentrated composed of 0% (Pea-0), 30% (Pea-30), or 60% (Pea-60) of pea.

	Treatments			SEM ^2^	*p*
	Pea-0	Pea-30	Pea-60		
Herbage mass, kg DM/ha					
Pre-grazing	2.330	2.393	2.383	79.10	0.80
Post-grazing	1.747	1.765	1.742	41.90	0.73
Intake, kg DM/cow/d					
Pasture	3.80	3.90	4.10	0.24	0.59
Grass silage	7.12	7.17	7.14	0.13	0.92
Concentrate	6.18	6.18	6.18	–	–
Total	17.1	17.3	17.3	0.24	0.70
Nutrient intake ^1^					
CP (kg CP/d)	2.83 ^b^	2.91 ^ab^	2.97 ^a^	0.04	0.04
ME (Mcal/d)	44.6	45.2	46.3	0.56	0.09
NDF (kg NDF/d)	6.6.25	6.24	6.08	0.10	0.38

^1^ Abbreviations: DM = dry matter; CP = crude protein; NDF = neutral detergent fiber; ME = metabolizable energy. ^2^ Standard error of the mean. – Not available values. ^a,b^ Values within a row with different superscripts differ significantly at *p* < 0.05.

**Table 5 animals-14-02830-t005:** Milk production and body weight of grazing dairy cows supplemented with concentrate composed of 0% (Pea-0), 30% (Pea-30), or 60% (Pea-60) pea.

	Treatments			SEM ^3^	*p*
	Pea-0	Pea-30	Pea-60		
Milk production (kg/d)	23.7	24.0	23.9	0.64	0.77
ECM ^1^ (kg/d)	26.9	27.7	27.3	0.62	0.51
FPCM ^2^ (kg/d)	25.4	25.7	25.7	0.60	0.74
Milk fat content (%)	4.36	4.52	4.43	0.16	0.65
Milk protein content (%)	3.30	3.26	3.25	0.07	0.68
Urea (mg/L)	285 ^b^	330 ^a^	335 ^a^	16.4	0.02
Milk Fat (kg/d)	1.0	1.1	1.1	0.03	0.40
Milk protein (kg/d)	0.78	0.78	0.77	0.02	0.94
Body weight, kg	641	638	642	7.50	0.42
Change in body weight (kg/d)	−0.202	−0.047	−0.155	0.15	0.80

Abbreviations: ^1^ Energy-corrected milk (ECM) = (12.97 × kg of fat) + (7.21 × kg of protein) + (0.3273 × kg of milk) [29]. ^2^ Fat- and protein-corrected milk = milk (kg/d) × (0.1226 × fat % + 0.0776 × protein % + 0.2534) [30]. ^3^ Standard error of the mean. ^a,b^ Values within a row with different superscripts differ significantly at *p* < 0.05.

**Table 6 animals-14-02830-t006:** Blood parameters of grazing dairy cows supplemented with concentrate composed of 0% (Pea-0), 30% (Pea-30), or 60% (Pea-60) of pea.

Blood Parameters ^1^	Treatments			SEM ^2^	*p*
	Pea-0	Pea-30	Pea-60		
BHB (mmol/L)	0.75 ^b^	0.74 ^b^	0.65 ^a^	0.06	0.03
Urea (mmol/L)	6.83	6.69	6.51	0.31	0.43
Albumin, g/L	38.2	39.2	39.2	1.67	0.75
Ca (mmol/L)	2.36	2.33	2.35	0.06	0.80
P (mmol/L)	1.63	1.61	1.68	0.09	0.76
Mg (mmol/L)	0.81	0.81	0.87	0.04	0.17
AST (u/L)	77.8	77.8	79.2	6.30	0.91

^1^ Abbreviations: BHB = β-hydroxybutyrate; AST = Aspartate Amino Transferase. ^2^ Standard error of the mean. ^a,b^ Values within a row with different superscripts differ significantly at *p* < 0.05.

**Table 7 animals-14-02830-t007:** Ruminal fermentation parameters of grazing dairy cows supplemented with concentrate composed of 0% (Pea-0), 30% (Pea-30), or 60% (Pea-60) of pea.

	Treatments			SEM	Time of Day		SEM ^2^	*p*		
	Pea-0	Pea-30	Pea-60		A.M	P.M		Trt	Time	Int
pH	6.72	6.73	6.80	0.06	6.71	6.79	0.05	0.45	0.15	0.41
VFA ^1^, mmol/100 mol										
Acetate	33.7	37.4	31.8	2.40	35.6	33.0	2.02	0.23	0.33	0.36
Butyrate	11.5	11.6	11.3	0.62	10.5	12.5	0.40	0.93	<0.01	0.67
Propionate	12.7	13.1	12.0	0.67	11.4	13.8	0.57	0.49	<0.01	0.37
Isobutyrate	2.11	2.12	2.04	0.09	2.14	2.04	0.08	0.82	0.28	0.60
Isovaleric	1.97	1.94	1.87	0.13	1.98	1.88	0.11	0.83	0.49	0.62
Valeric	1.62	1.56	1.52	0.05	1.52	1.62	0.04	0.51	0.17	0.67
Total, mmol/L	63.6	67.8	60.6	3.24	63.2	64.8	2.70	0.30	0.65	0.37

^1^ Abbreviations: VFA = volatile fatty acids. ^2^ Standard error of the mean.

## Data Availability

The original contributions presented in the study are included in the article, further inquiries can be directed to the corresponding author.

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
