# Peer review of "Effect of Replacing Corn Grain and Soybean Meal with Field Peas at Different Levels on Feed Intake, Milk Production, and Metabolism in Dairy Cows under a Restrictive Grazing"

_animals, 2024, doi:10.3390/ani14192830_

Round 1

Reviewer 1 Report (Previous Reviewer 1)

Comments and Suggestions for Authors

Dear Authors, 

I have reviewed the revised version of manuscript animals-3225746. Compared to the first submission, the manuscript has been greatly improved and deserves acceptance in the journal Animals. However, I could not help but note a few minor details, which I list below:

- L 19: In my opinion, the phrase "like summer" is redundant and unnecessary. You may consider deleting it;

- L 29: Please check the spelling of "ruminal", thanks;

- L 360 (and throughout the text): I suggest you avoid starting a sentence with an acronym, thanks;

- L 375: Reference [19] should be cited as Vander Pol et al. (check for yourself at https://www.sciencedirect.com/science/article/pii/S0022030208714139). For reference [36], you should include the name of the first author in the text (i.e., Tufarelli et al. [36]).

L 378: In my opinion, you could also briefly mention what has been found in other dairy species. This would make your results more robust. For sheep and buffalo, the following references may be valid https://doi.org/10.1071/AN14019 and https://doi.org/10.1007/s10457-018-0316-5

Author Response

Response to Reviewer 1 comments:

Dear reviewer, thank you for your constructive feedback. We have made the suggested revisions.

- L 19: In my opinion, the phrase "like summer" is redundant and unnecessary. You may consider deleting it.

R: We have removed the phrase "like summer" to improve the clarity of the sentence.

- L 29: Please check the spelling of "ruminal", thanks.

R: We have reviewed and corrected the spelling of "ruminal" in L 29 and throughout the manuscript where necessary.

- L 360 (and throughout the text): I suggest you avoid starting a sentence with an acronym, thanks.

R: We have revised the manuscript to ensure that sentences no longer begin with acronyms, improving readability and consistency.

- L 375: Reference [19] should be cited as Vander Pol et al. (check for yourself at https://www.sciencedirect.com/science/article/pii/S0022030208714139). For reference [36], you should include the name of the first author in the text (i.e., Tufarelli et al. [36]).

R: We have updated the citation for reference [19] to follow the correct format.

L 378: In my opinion, you could also briefly mention what has been found in other dairy species. This would make your results more robust. For sheep and buffalo, the following references may be valid https://doi.org/10.1071/AN14019 and https://doi.org/10.1007/s10457-018-0316-5

R: We have included a brief mention of findings in other dairy species, such as sheep and buffalo, as suggested.

Reviewer 2 Report (Previous Reviewer 2)

Comments and Suggestions for Authors

This article is a great improvement over previous versions, but the authenticity of the data in Table 2 is questionable. Please explain why the data is different from the original table.

Author Response

Dear reviewer, thank you for your constructive feedback. We have made the suggested revisions.

This article is a great improvement over previous versions, but the authenticity of the data in Table 2 is questionable. Please explain why the data is different from the original table.

R: Dear reviewer, thank you for your observation. The discrepancy in Table 2 was due to an error when initially presenting the information, as different individuals prepared the data, leading to inconsistencies. After the first round of reviews and thanks to your corrections, we realized this mistake. We then requested updated data from the company that produced the concentrate, and the table was corrected accordingly. We apologize for not clearly communicating this change in the previous revision.

Round 2

Reviewer 2 Report (Previous Reviewer 2)

Comments and Suggestions for Authors

can be accept in the present form.

This manuscript is a resubmission of an earlier submission. The following is a list of the peer review reports and author responses from that submission.

Round 1

Reviewer 1 Report

Comments and Suggestions for Authors

Manuscript ID animals-2418631

Effect of replacing corn grain and soybean meal with field peas at different levels on feed intake, milk production and metabolism in dairy cows under a restrictive grazing

GENERAL REMARKS

The search for alternative protein sources to soy products in animal nutrition represents a topic of extreme economic and social importance, in which your research fits. Although extensively addressed, the use of peas in the diet of dairy cows in association with pasture is a little-explored option, so I find your manuscript very useful in completing my knowledge in this area. Coming to the manuscript, I find the introduction well focused on the problem and the results well described. Part of the materials and methods would probably need to be improved. Furthermore, as a subject much studied, some other studies (in part I have indicated them) should be recalled in the discussions in my opinion. My punctual comments are listed below, point by point. I wish I could contribute to the improvement of the manuscript.

SPECIFIC COMMENTS

L 20 (and in other parts of the text, as at L 22, 34, 37, 70): as per the journal’s template, the acronyms should be specified at first mention. In addition, both in the simple summary and abstract, I think it more appropriate to avoid the use of acronyms since these manuscript's parts are also intended for the lay public, which may have difficulty contextualizing the contents. Thanks.

L 69 (and in other parts of the text, as at L 110): botanical names of plant species should be in italics. Thanks.

L 80: according to the style used for the other references listed through the text, the reference referring to Albrecht should be followed by the sequential reference number rather than the year of publication. Thanks.

L 107: I think it is more appropriate to use "blocked" instead of "matched". Thanks.

L 120: literally, the "individual feed pens" should refer to individual feed boxes in which each cow should be individually confined. However, in the period below (L 121-122) I read that water is fully available in the paddock and cowshed, leading me to assume that the animals were housed in groups. To avoid misinterpretations, I ask you if it would be possible to better detail the housing conditions of the experimental animals. Thanks.

L 125: according to the table heading, the contents of the table should be inverted. In addition, do the nutritional characteristics of whole diets include the proportion of pasture? In my opinion, this issue is not univocally stated. Also, I think it useful to specify the CP contents of soybean meal and pea grains used in the formulation of the experimental concentrates. Thanks.

L 143: More than 50 m wide, I suggest you indicate the overall dimensions of the grazing strips. In this way, we do not have an account of the effective capacity of the pasture in relation to the load of livestock that can be maintained. Thanks.

L 165: are only 2 cows grazed? This statement contradicts the previously specified number of cows per group (n = 4, see L 109).

L 166: if the silage residues and concentrates were weighed individually, I assume they were not fed as a total mixed ration. However, this contradicts what was stated in lines 120-121. Please check, thanks.

L 210 (and through the text): according to the journal’s template, the “p-value” should be reported in lowercase and italic. Thanks.

L 216-218: I think it's a template typo. Please delete this part, thanks.

L 237: for silage only the SEM is reported and not the p-value, please check. Thanks.

L 254: The table shows (first column) the chemical composition, which, however, does not seem to me to be specified in the table. I suggest you check it out, thanks.

L 267: Let's see milk corrected?

L 287: I think “BHB” was not previously defined as acronymous. Please check, thanks.

L 310: I find the discussions well-structured and sufficiently argued. However, in addition to those indicated, there are also other studies that the authors could use to support their findings, including, for example:

-        https://doi.org/10.5194/aab-55-132-2012,

-        https:// doi.org/10.1111/j.1740-0929.2011.00934.x,

-        https://doi.org/10.4141/cjas95-092

-        https://doi.org/10.4081/ijas.2006.237.

Furthermore, I would like to advise the authors not to narrow their field of vision exclusively to the dairy cow. Other authors, in fact, have evaluated the effect of administering concentrates formulated with peas instead of soy in large dairy species, such as buffalo (https://doi.org/10.1007/s10457-018-0316 -5). I believe that recalling these results could be useful to give more weight to the results proposed in this manuscript and highlight the validity of the alternative protein source studied. Thank you.

L 324-329: From your point of view, are the observed differences in CP content attributable to the aging of the vegetation that occurred during the test or are they intrinsically linked to the grass grazed by the groups? Since the grass has been of a single species and managed equally, I don't think it's easy to understand these differences, the causes of which it would perhaps be useful to add something, in my opinion. I ask for your opinion on this, thank you.

Author Response

Response to Reviewer 1 comments:

L 20 (and in other parts of the text, as at L 22, 34, 37, 70): as per the journal’s template, the acronyms should be specified at first mention. In addition, both in the simple summary and abstract, I think it more appropriate to avoid the use of acronyms since these manuscript's parts are also intended for the lay public, which may have difficulty contextualizing the contents. Thanks.

R: We have revised the manuscript to define all acronyms at their first use. Additionally, we reduced the use of acronyms in the abstract and simple summary to make these sections more accessible.

L 69 (and in other parts of the text, as at L 110): botanical names of plant species should be in italics. Thanks.

R: We have corrected the formatting in the manuscript to ensure that all botanical names are italicized as per standard scientific convention.

L 80: according to the style used for the other references listed through the text, the reference referring to Albrecht should be followed by the sequential reference number rather than the year of publication. Thanks.

R: We have corrected the citation style for Albrecht and ensured that all references throughout the manuscript now conform to the sequential reference number format as required by the journal’s guidelines.

L 107: I think it is more appropriate to use "blocked" instead of "matched". Thanks.

R: Corrected.

L 120: literally, the "individual feed pens" should refer to individual feed boxes in which each cow should be individually confined. However, in the period below (L 121-122) I read that water is fully available in the paddock and cowshed, leading me to assume that the animals were housed in groups. To avoid misinterpretations, I ask you if it would be possible to better detail the housing conditions of the experimental animals. Thanks.

R:  We have revised the manuscript to include more detailed information about the experimental facilities, ensuring clear understanding of the housing setup for the animals.

L 125: according to the table heading, the contents of the table should be inverted. In addition, do the nutritional characteristics of whole diets include the proportion of pasture? In my opinion, this issue is not univocally stated. Also, I think it useful to specify the CP contents of soybean meal and pea grains used in the formulation of the experimental concentrates. Thanks.

R: We have updated all tables in the manuscript to ensure clarity and accuracy of information. These revisions include presenting the detailed composition of the ingredients in the concentrates.

L 143: More than 50 m wide, I suggest you indicate the overall dimensions of the grazing strips. In this way, we do not have an account of the effective capacity of the pasture in relation to the load of livestock that can be maintained. Thanks.

R: We have revised the manuscript to include more detailed information about the grazing management and the specific dimensions of the grazing strips in the methodology section.

L 165: are only 2 cows grazed? This statement contradicts the previously specified number of cows per group (n = 4, see L 109).

R: We have rewritten this paragraph to clarify the methodology foe determining pasture DMI in grazing cows.

L 166: if the silage residues and concentrates were weighed individually, I assume they were not fed as a total mixed ration. However, this contradicts what was stated in lines 120-121. Please check, thanks.

R: We have corrected the description to clarify that the amounts offered and the refusals of the PMR were calculated as a whole, not individually for each ingredient.

L 210 (and through the text): according to the journal’s template, the “p-value” should be reported in lowercase and italic. Thanks.

R: We have revised the manuscript to ensure that all instances of “p-value” are reported in lowercase and italic throughout the text, in accordance with the journal's formatting guidelines.

L 216-218: I think it's a template typo. Please delete this part, thanks.

R: Corrected.

L 237: for silage only the SEM is reported and not the p-value, please check. Thanks.

R: This was an error. The value reported was the standard deviation (SD), not the SEM. We have updated the Table to correctly reflect this.

L 254: The table shows (first column) the chemical composition, which, however, does not seem to me to be specified in the table. I suggest you check it out, thanks.

R: We have updated and corrected Table 4

L 267: Let's see milk corrected?

R: This line was removed.

L 287: I think “BHB” was not previously defined as acronymous. Please check, thanks.

Corrected.

L 310: I find the discussions well-structured and sufficiently argued. However, in addition to those indicated, there are also other studies that the authors could use to support their findings

R: We appreciate the valuable suggestions. We have included some of the recommended references to support and enrich the manuscript discussion.

L 324-329: From your point of view, are the observed differences in CP content attributable to the aging of the vegetation that occurred during the test or are they intrinsically linked to the grass grazed by the groups? Since the grass has been of a single species and managed equally, I don't think it's easy to understand these differences, the causes of which it would perhaps be useful to add something, in my opinion. I ask for your opinion on this, thank you.

R: The observed differences in CP content were due to variations in pasture quality related to the type of grazing management employed. Specifically, the quality of the pasture differed slightly due to the treatments grazing through the paddocks in a consistent order throughout the study. This clarification has been added to the discussion to explain the underlying reasons for the differences observed in CP content.

Reviewer 2 Report

Comments and Suggestions for Authors

Pulido et al. conducted this study to determine the effect on milk production, intake, rumen fermentation and blood indicators of substituting soybean meal and corn grain for raw peas in the 27 concentrate of grazing dairy cows. The topic selection is very meaningful, but there are some several weaknesses in the present manuscript.

(1) the authors did not show the nutrient composition of pea before used in the experimental diets. As the authors point out, there are few field studies using peas as the primary source of protein and starch for dairy cows, particularly in grazing cows. The authors should analyze its nutrient composition, especially crude protein, crude fat, crude fiber, etc., because the use of peas in concentrate in this experiment is very large.

(2) It is not stated what ingredient in concentrate was replaced by peas in the experimental diet.

(3) The design of concentrate protein level of experimental diet was not reasonable, the protein level of experimental group was much higher than that of control group (Pea-30 group was 0.7% higher than that of control group, and Pea-60 group was even 1% higher).

(4) table 2, the crude protein of pasture offered to grazing dairy cows during the experiment in each group were inconsistent, which may affect the results of subsequent experiments to some extent, the same as in table 3.

(5) table 4, the crude protein intake of experimental cows was significantly higher than that of control cows. This is most likely due to higher protein levels in the experimental group than in the control group.

(6) in the abstract, the authors say that “Milk production, milk composition and ruminal metabolites did not differ among treatments.”, however, it can be clearly seen in Table 5 that urea in milk in the experimental group was significantly higher than that in the control group.

Author Response

Response to Reviewer 2 comments:

(1) the authors did not show the nutrient composition of pea before used in the experimental diets. As the authors point out, there are few field studies using peas as the primary source of protein and starch for dairy cows, particularly in grazing cows. The authors should analyze its nutrient composition, especially crude protein, crude fat, crude fiber, etc., because the use of peas in concentrate in this experiment is very large.

R: We have included now a table in the manuscript that presents the detailed nutrient composition of the peas used in the experimental diets.

(2) It is not stated what ingredient in concentrate was replaced by peas in the experimental diet.

R: We have updated Table 2 to specify the ingredients in the concentrate that were replaced by peas in the experimental diet. This provides clear information on the formulation changes made for the experimental diets.

(3) The design of concentrate protein level of experimental diet was not reasonable, the protein level of experimental group was much higher than that of control group (Pea-30 group was 0.7% higher than that of control group, and Pea-60 group was even 1% higher).

R: We acknowledge this concern. We have revised the manuscript to clarify that the experimental diets were designed to be isonitrogenous and isoenergetic, ensuring that protein levels were balanced across all groups. This adjustment has been detailed in the methodology and discussed in the results section to explain the rationale and the actual implementation of the diet formulations.

(4) table 2, the crude protein of pasture offered to grazing dairy cows during the experiment in each group were inconsistent, which may affect the results of subsequent experiments to some extent, the same as in table 3.

R: The observed differences in CP content were due to variations in pasture quality related to the type of grazing management employed. Specifically, the quality of the pasture differed slightly due to the treatments grazing through the paddocks in a consistent order throughout the study. This clarification has been added to the discussion to explain the underlying reasons for the differences observed in CP content.

(5) table 4, the crude protein intake of experimental cows was significantly higher than that of control cows. This is most likely due to higher protein levels in the experimental group than in the control group.

R: We acknowledge this situation. Part of the discussion section was rewritten to address this issue and align with the results, and its implications have been detailed in the discussion section of the manuscript.

(6) in the abstract, the authors say that “Milk production, milk composition and ruminal metabolites did not differ among treatments.”, however, it can be clearly seen in Table 5 that urea in milk in the experimental group was significantly higher than that in the control group.

R: We have revised and rewritten the abstract to accurately reflect the findings, including the significant difference in urea levels in milk between the experimental and control groups.